# Protein Phosphorylation Alterations in Myotonic Dystrophy Type 1: A Systematic Review

**DOI:** 10.3390/ijms24043091

**Published:** 2023-02-04

**Authors:** Adriana Costa, Ana C. Cruz, Filipa Martins, Sandra Rebelo

**Affiliations:** Institute of Biomedicine (iBiMED), Department of Medical Sciences, University of Aveiro, 3810-193 Aveiro, Portugal

**Keywords:** protein kinases, protein phosphatases, phosphoproteins, protein phosphorylation, human samples, animal and cellular models, myotonic dystrophy type 1

## Abstract

Among the most common muscular dystrophies in adults is Myotonic Dystrophy type 1 (DM1), an autosomal dominant disorder characterized by myotonia, muscle wasting and weakness, and multisystemic dysfunctions. This disorder is caused by an abnormal expansion of the CTG triplet at the *DMPK* gene that, when transcribed to expanded mRNA, can lead to RNA toxic gain of function, alternative splicing impairments, and dysfunction of different signaling pathways, many regulated by protein phosphorylation. In order to deeply characterize the protein phosphorylation alterations in DM1, a systematic review was conducted through PubMed and Web of Science databases. From a total of 962 articles screened, 41 were included for qualitative analysis, where we retrieved information about total and phosphorylated levels of protein kinases, protein phosphatases, and phosphoproteins in DM1 human samples and animal and cell models. Twenty-nine kinases, 3 phosphatases, and 17 phosphoproteins were reported altered in DM1. Signaling pathways that regulate cell functions such as glucose metabolism, cell cycle, myogenesis, and apoptosis were impaired, as seen by significant alterations to pathways such as AKT/mTOR, MEK/ERK, PKC/CUGBP1, AMPK, and others in DM1 samples. This explains the complexity of DM1 and its different manifestations and symptoms, such as increased insulin resistance and cancer risk. Further studies can be done to complement and explore in detail specific pathways and how their regulation is altered in DM1, to find what key phosphorylation alterations are responsible for these manifestations, and ultimately to find therapeutic targets for future treatments.

## 1. Introduction

Myotonic Dystrophy type 1 (MIM # 160900), also known as Steinert’s disease, is one of the most common muscular dystrophies in adults, with an incidence of 8.3 to 10.6 in 100,000 people [1,2]. DM1 is mainly characterized by myotonia, progressive muscle wasting and weakness, as well as variable multisystemic features, such as insulin resistance, dyslipidemia, cardiac conduction defects, respiratory dysfunctions, gonadal atrophy, cataracts, and central nervous system alterations [3,4,5,6]. Of note, the cardiac conduction abnormalities and respiratory dysfunctions are highly prevalent and are the main causes of mortality in patients with DM1 [7].

DM1 is caused by a mutation leading to an expansion of the CTG triplet at the 3′UTR region of the *Dystrophia Myotonica Protein Kinase* (*DMPK*) gene (MIM * 605377), located at chromosome 19q13.32 [8]. Among patients with DM1, CTG expansions can range from 50 to more than 2000 repeats, with increased severity and earlier age of onset of symptoms in the higher CTG repeat lengths. The major hypothesis explaining DM1 pathophysiology is the RNA gain of toxic function, being currently the one presenting the strongest evidence [9]. Of note, in the expanded sequence of CTG nucleotides in the *DMPK* gene, several CUG expanded mRNAs are transcribed. These expanded mRNAs fold into hairpin-like structures that accumulate in the nucleus, trapping the RNA splicing factors, such as muscleblind-like splicing regulator 1 (MBNL1) and Heterogeneous nuclear ribonucleoprotein H (hnRNP H), leading to their depletion and loss of function. On the other hand, another RNA splicing factor named CUG binding protein 1 (CUGBP1) does not bind to these expanded mRNAs and is activated through hyperphosphorylation [10]. This subsequently impairs the expression and splicing of different mRNAs and proteins [10], affecting several signaling pathways and the phosphorylation levels of intervenient, being one of the most common post-translational modifications [11].

Reversible protein phosphorylation at serine (Ser), threonine (Thr), and tyrosine (Tyr) residues is among the major regulatory mechanisms in eukaryotic cells, controlling key intracellular events that are essential for cell health and viability [11,12]. Protein kinases are responsible for the mechanism of phosphorylation, adding phosphate groups and activating, or inhibiting, the activity of different proteins, therefore regulating many cellular pathways and events [11]. On the other hand, protein phosphatases are responsible for removing the phosphate group from the different phosphoproteins, which will return to an unphosphorylated state. Imbalanced kinase/phosphatase activities impair signal transduction pathways and are associated with several diseases, such as Alzheimer’s disease and cancer [13,14,15,16,17,18].

Despite substantial progress made in the understanding of DM1, the identification of molecular mechanisms underlying this pathology is still elusive. Of note, DMPK is a protein kinase, and several substrates whose altered phosphorylation levels are directly associated to DM1 have been identified. Among these DMPK substrates are phospholamban (PLB), phospholemman (PLM), and CUGBP1 [19]. Other phosphoproteins, including sarcolipin and lipin, have been implicated in cellular events relevant for DM1 pathophysiology, namely Ca^2+^ homeostasis and lipid metabolism, which are associated to myotonia and insulin resistance, respectively [6].

Some evidence suggests that certain proteins have altered phosphorylation and expression levels in DM1 [20]. For example, CUG toxic gain of function is thought to activate the protein kinase C (PKC) signaling pathway, leading to the hyperphosphorylation and upregulation of the CUGBP1 steady-state levels, contributing to its nuclear stabilization [9]. However, the magnitude of how phosphorylation is impaired in the different signaling pathways is not well established. Unraveling the mysteries of phosphorylation imbalances associated to DM1—namely the protein kinase and phosphatase levels and activity alterations—could represent a powerful strategy to implement new therapies and improve symptoms and overall quality of life of patients with DM1.

The aim of this systematic review was to gather data concerning protein phosphorylation in DM1, understand which kinases, phosphatases, and phosphoproteins have altered expression, phosphorylation levels and activity, and highlight the most impaired signaling pathways regulated by protein phosphorylation associated to DM1.

## 2. Methods

This systematic review was performed according to the Preferred Reporting Items for Systematic Reviews and Meta-Analysis (PRISMA) checklist for systematic reviews and meta-analysis. The protocol used for this literature review was firstly searched on the international prospective registry for systematic reviews (PROSPERO) [21,22] and submitted on the platform after excluding the existence of reviews or protocols with the same purpose. The protocol used for this review was registered in PROSPERO (CRD420211186960).

### 2.1. Search Strategy

The literature searches were conducted in the PubMed and Web of Science databases and were complemented by weekly automatic updates retrieved up until 10 May 2022. Both a keyword search and PubMed Mesh term search were performed, including keywords/terms such as “Myotonic Dystrophy”, “protein phosphorylation”, “proteomics”, and “phosphoproteins”. The full search strategy is described as Appendix A

The studies were considered eligible if they (1) included animal and/or cellular models of DM1; (2) included samples from patients with DM1; (3) were written in English or Portuguese. Studies were excluded if (1) the abstract and title were not available; (2) they were abstracts/conference abstracts, books, book chapters, case reports, commentaries, dissertations, editorials, guidelines, letters to the editor, meta-analysis, news, qualitative studies, position papers, research protocols, statements, systematic reviews, theses, and unpublished work; (3) DM1 was not addressed; (4) protein phosphorylation was not evaluated.

One independent researcher checked for duplicated studies, screened the studies’ titles and abstracts, and subsequently reviewed the full text of the selected records for qualitative and quantitative analysis. In case of doubt, a second researcher was consulted for further data analysis.

### 2.2. Data Extraction, Synthesis and Analysis

Upon selection of the final studies, the data was extracted to a well-structured table containing author, year and country of the publication, sample type and characterization (e.g., sample size, CTG repeat length, tissue, and age), techniques used for protein evaluation, and the phosphorylation-associated proteins and their main findings (total protein level and phosphorylated levels) (Appendix A). A summary table was constructed to summarize the alterations observed in protein phosphorylation associated to DM1. In overall, the data was organized by protein (kinases, phosphatases, and phosphoproteins) and the total, phosphorylated, and activity levels of each protein was presented, comparing DM1 human, animal and cell line samples with corresponding control samples (DM1 vs. CTL). Results were considered similar when the difference between DM1 and CTL was not statistically significant. *p*-values are depicted as “p: NR” when they were not reported in the studies.

Data was extracted by one researcher and verified by a second and third researcher for accuracy and completeness of data. Discrepancies were solved first by consensus, and if consensus was still not possible, a fourth reviewer was consulted.

BioRender, a web-based tool with thousands of premade icons was used to create scientific figures. 

## 3. Results and Discussion

From the PubMed and Web of Science database search, 1623 studies were retrieved. After 661 duplicates were removed, 962 studies were screened by title and abstract in the exclusion criteria analysis, in which 490 were excluded: 263 for the type of study, 107 were not focused on molecular research, 43 studies were not related to DM1, and 77 were excluded by language, which resulted in 472 studies accepted for a second screening with respect to the scientific content of the abstract (full texts were analyzed when necessary). From the 472 studies, 244 did not evaluate proteins and one was a review. Further, 227 studies remained for the full-text analysis, in which 181 did not present relevant data for the aim of this review and four did not use DM1 biological models. Finally, 41 studies [23,24,25,26,27,28,29,30,31,32,33,34,35,36,37,38,39,40,41,42,43,44,45,46,47,48,49,50,51,52,53,54,55,56,57,58,59,60,61,62,63] were included in the present systematic review. The details of the screening process can be found in the PRISMA flow diagram in Figure 1.

### 3.1. Characterization of Sample Type Used in the Included Studies

Regarding the 41 studies included in present systematic review, the human biological samples were the most used for the evaluation of protein phosphorylation alterations in patients with DM1, followed by use of mice model derived samples (Figure 2). Additionally, six studies used cell lines, namely induced pluripotent stem cell (IPSC)-derived satellite skeletal muscle cells, Chinese hamster ovary cell (CHO) cells, Cellosaurus cell line M6 (COS M6), C2C12 cell line, and PC12 cell line [31,37,51,56,60,63]. Interestingly, the tissues of choice were the skeletal muscle (n = 25 for humans; n = 10 for mice models) and heart (n = 1 for humans; n = 4 for mice models) (Figure 2A). These are important tissues given that patients with DM1 frequently present muscle impairment and cardiac conduction dysfunctions [64]. Transgenic mice expressing the human skeletal actin gene with 250 CTG repeats (HSALR) was the most used mice model to study protein phosphorylation in DM1 (n = 4) [26,33,43,44] (Figure 2B). Further, the DMSXL mice (n = 3) [53,54,55], *Dmpk* knockout mice (n = 2) [38,39], and inducible transgenic mice with expanded repeats induced by Tamoxifen (TAM) (n = 2) [37,52] were also among the most used DM1 mice models (Figure 2B). Concerning the human samples, in most studies skeletal muscle biopsies were used (n = 15) [26,28,29,32,33,34,40,45,46,50,57,58,59,61,62], followed by primary myoblasts (n = 7) [24,31,33,47,49,50,53] and primary myotubes (n = 4) [25,28,47,49] (Figure 2C).

### 3.2. Protein Kinases

Among the 41 studies, the total protein levels as well as protein phosphorylation levels of 29 kinases were analyzed (Figure 3A; Table 1). The most evaluated kinase was protein kinase B (AKT), which was evaluated in 10 studies [26,27,30,35,38,45,46,47,48,49], followed by DMPK [24,28,30,50,56,57,58,59,62] evaluated in nine studies and glycogen synthase kinase 3β (GSK3β) [27,29,33,38,45,53] evaluated in six studies. Both 5′ AMP-activated protein kinase (AMPK) [26,27,40,44,48] and ribosomal protein S6 kinase β-1 (S6K1) [25,27,46,47,48] were evaluated in five studies. Extracellular signal-regulated kinases 1 and 2 (ERK1 and ERK2) [24,35,45,46] were evaluated by four studies, mechanistic target of rapamycin (mTOR) [26,46,51] and protein kinase C α/βII (PKCα/βII) [37,51,52] were evaluated by three studies, while protein kinase R (PKR) [31,41] and p38 mitogen-activated protein kinases (p38MAPK) [24,30] and CDK4 [49,61] were evaluated by two studies. All the remaining kinases were evaluated in one study (Figure 3A; Table 1).

### 3.3. Protein Phosphatases

Phosphatases were the class of proteins less assessed across the studies, with three phosphatases retrieved that were analyzed by one article each: Type II inositol 3,4-bisphosphate 4-phosphatase (INPP4B) [23], Receptor-type tyrosine-protein phosphatase F (PTPRF) [23], and Calcineurin [43] (Figure 3B; Table 1).

### 3.4. Phosphoproteins

The phosphorylation status of 17 phosphoproteins was evaluated across the 41 included studies (Figure 3C, Table 1). The most evaluated phosphoproteins were the RNA binding protein CUGBP1 [31,33,36,37,49,52,53] reported in seven studies and Tau protein [42,55,63] reported in three studies, followed by Retinoblastoma protein (Rb) [25,28], Ribosomal protein S6 (RpS6) [26,27], Eukaryotic translation eIF2α [31,32], Insulin Receptor Substrate 1 (IRS1) [45,46], and c-Jun [40,60] reported in two studies (Figure 3C). All the remaining phosphoproteins were only evaluated by one study each (Figure 3C).

### 3.5. Deregulation of Major Signalling Pathways Associated to DM1 Regulated by Protein Phosphorylation

#### 3.5.1. AKT/mTOR Pathway

AKT is a serine/threonine kinase that promotes cell survival, proliferation, and glucose uptake and synthesis by interacting and phosphorylating targets such as GSK3β, mTOR, AS160, and FOXO1 [65]. In DM1, AKT total levels were quite similar to control in most of the reported biological samples (Appendix A, Table 1), but were found increased in human DM1 tibialis anterior (TA) skeletal muscle biopsies (*p* = 0.02) [45] (Appendix A; Table 1) and decreased in human DM1-derived congenital myoblasts with 3200 CTG repeats (*p* < 0.05) [47] (Appendix A; Table 1). The dual phosphorylation of AKT at Thr308 and Ser473 residues is responsible for its activation. Phosphorylated AKT (p-AKT) levels at Thr308 were increased in DM1 human distal skeletal muscle biopsies (*p* = 0.04), while no difference was detected in proximal skeletal muscle biopsies [45] (Appendix A; Table 1). Phosphorylation levels at Ser473 differed between human myoblasts and myotubes, where they were increased and decreased compared to control, respectively [48,49] (Appendix A, Table 1). Also, p-AKT levels were decreased in DM1 human fibroblasts [30] (Table 1).

GSK3 is a multi-functional serine/threonine kinase that inhibits the glycogen synthase activity, involved in cell proliferation, metabolism, and cell survival [65]. There are two isoforms, GSK3β and GSK3α, which are constitutively active, and are inhibited upon AKT phosphorylation at Ser9 and Ser21, respectively, favoring glycogen formation and storage [65]. GSK3β total levels in DM1 human skeletal muscle biopsies varied according to the biopsy site (Appendix A and Table 1), since they were significantly decreased in human TA distal muscle biopsies (*p* = 0.03) [45] but increased in BB proximal muscle biopsies [33]. Further, GSK3β total levels were also increased in DM1 human myoblasts [53] (Appendix A and Table 1) and mice skeletal muscle biopsies [33,36,53] (Appendix A and Table 1). Regarding phosphorylation of GSK3, there is consensus that phosphorylated GSK3β at Ser9 (inhibited state) is decreased in DM1 human skeletal muscle biopsies when compared to control [33] and in neural stem cells (*p* < 0.05) [27] (Appendix A and Table 1). Phosphorylated GSK3β at Tyr216 (active state) was increased in human distal (*p* = 0.004) [45] and proximal muscle biopsies [33] (Appendix A and Table 1). Lastly, GSK3α phosphorylated levels at Ser21 (inhibited state) were significantly decreased in human neural stem cells in patients with DM1 (*p* < 0.05) [27] (Table 1). Overall, these results indicate that phosphorylated and inhibited forms of GSK3 are decreased, suggesting an increase of GSK3 activity, possibly due to a decrease in phosphorylated levels of AKT.

mTOR is a serine/threonine kinase that stimulates muscle protein synthesis and cell growth [65]. AKT is suggested to phosphorylate and directly activate mTOR at Ser2448 and indirectly through the inhibition of TSC2, which is a mTOR inhibitor [65]. In DM1, phosphorylated mTOR levels at Ser2448 were decreased in human- and IPSC-derived satellite skeletal muscle cells (*p* < 0.05) [51] (Table 1), suggesting that mTOR activity is decreased in DM1. Further, mTOR phosphorylates and activates S6K1 at Thr389, which promotes translation of cell proliferation-related mRNA [66]. In DM1, the phosphorylated S6K1 levels at Thr389 were increased in DM1 human skeletal muscle biopsies and DM1 mice skeletal muscle biopsies upon 24 h starvation compared to control [26] (Appendix A and Table 1). Also, mTOR protein complex 1 (mTORC1) and c-Jun N-terminal kinase (JNK) phosphorylates S6K1 at Thr421/Ser424, which disables the autoinhibition domain, inducing a S6K1 conformational change that facilitates its phosphorylation and activation [67]. S6K1 phosphorylated levels at these two residues were significantly decreased before and during myoblasts to myotubes differentiation [24] (Appendix A, Table 1).

RpS6 is a target of S6K1, which is phosphorylated and activated at Ser235/Ser236/Ser240/Ser244 and is involved in translation [68]. In human skeletal muscle biopsies, phosphorylated RpS6 levels were increased [26] (Appendix A and Table 1). However, in human neural stem cells, phosphorylated RpS6 levels were decreased in DM1 when compared with controls [27] (Table 1).

In addition, three studies have evaluated the AKT/mTOR pathway upon insulin stimulation, resulting in its activation. Essentially, in these studies, a range of concentration between 10 nM and 100 nM of insulin was administrated to mice/cells for different periods of time, namely 0, 5, 15, 20 and 30 min. Results showed that this pathway is under stimulated in DM1 [38,45,46,69].

#### 3.5.2. AMPK Pathway

Another important signaling pathway evaluated in DM1 was the AMPK pathway, that functions as cell growth inhibitor, metabolism regulator, and is also associated to autophagy and cell polarity [70]. The kinase CaMKIIβ is an upstream effector of AMPK, that phosphorylates and activates AMPK at Thr172 residue upon an increase in calcium influx. In DM1 mice muscle tissue, there was a significant decrease in p-CaMKIIβM (Thr286) levels (*p* < 0.05) [26] (Appendix A and Table 1). Concerning total AMPK levels, they appear to be similar between DM1 and controls in almost all types of biological samples tested except for muscle biopsies from TREDT960I mice with severe muscle wasting, where they were found increased (*p* = 0.05) [40] (Appendix A and Table 1). The levels of phosphorylated AMPK at Thr172 were also similar between DM1 and controls in most models tested. However, there was a significant decrease of phosphorylated AMPK at this residue in muscle biopsies from HSALR mice (*p* = 0.02) [44] and in human fibroblast-derived myoblasts (*p* = 0.02) [44] (Appendix A and Table 1), while an increase was found in skeletal muscle biopsies obtained from TREDT960I mice with severe muscle wasting (*p* < 0.01) [40] (Appendix A and Table 1).

There were several alterations in the AMPK signaling pathway upon starvation in DM1 samples. Skeletal muscle biopsies from HSALR DM1 mice were collected and analyzed upon a period of starvation. These DM1 mice models did not respond to fasting conditions as compared to control [26]. Firstly, AMPK regulatory kinases LKB1 and TAK1 total levels were similar in both DM1 and control [26] (Table 1). However, a decrease was reported in total and phosphorylated active CaMKII (Thr286) levels in these mice models when compared to controls upon 24 h starvation (*p* < 0.05) [26] (Table 1). AMPK phosphorylation levels were not responsive upon 24 h of starvation, and reduced levels of p-AMPK (Thr172) in HSALR DM1 mice skeletal muscle biopsies were observed, while a significant increase was observed in control muscle biopsies (*p* < 0.05) [26]. However, p-AMPK levels were normalized after 45 h of starvation [26] (Table 1). Similar phosphorylation levels of the AMPK downstream targets TSC2 (Ser1387) and ULK1 (Ser317) were observed between DM1 and controls in 24 h starved mice [26] (Table 1).

#### 3.5.3. CUGBP1 Regulation

CUGBP1 is an RNA-binding protein, which regulates alternative splicing, mRNA degradation, and translation affecting gene expression. CUGBP1 total levels are commonly increased in DM1 human samples, mice, and cell models compared to controls [31,33,36,37] (Table 1). CUGBP1 is regulated by different proteins and protein complexes, which alters its affinity to different mRNAs [49]. CUGBP1 is phosphorylated at Ser302 and activated by the cyclinD3/CDK4/6 protein complex. This activation leads to the formation of an active translational complex with the unphosphorylated active form of eIF2α, which will increase the translation of mRNAs involved in DNA damage repair and chromatin remodeling [31]. In DM1, there was a decreased formation of these active translational complexes, since there is decreased interaction between CUGBP1 with cyclinD3 in myotubes and decreased levels of p-CUGBP1 at Ser302 in human myoblasts [31] and myotubes [49] (Appendix A, Table 1). Further, increased levels of inactive form of eIF2α, phosphorylated at Ser51, were detected in DM1 human myoblasts and CHO cell lines expressing expanded CUG RNA [31] (Appendix A and Table 1).

CUGBP1 can be phosphorylated by AKT at Ser28. Increased levels of p-CUGBP1 at Ser28 were detected in DM1 human myoblasts, which increased the interaction of CUGBP1 with cyclin D1 mRNA, a strong promoter of cell proliferation [49] (Appendix A and Table 1). Another regulator of CUGBP1 is PKC, which has increased phosphorylated levels in DM1 mice heart and muscle tissue, human heart tissue, and COS M6 cell lines (Table 1). In the studies that evaluated CUGBP1 and PKC, total and phosphorylated levels were both found to be increased [36,37,52,53]. CUGBP1 is considered a target of DMPK; however, no studies have evaluated both proteins simultaneously to understand how their interaction may be impaired in DM1. Therefore, this issue should be explored in future studies.

#### 3.5.4. MEK/ERK Pathway

The MEK/ERK pathway plays an important role in regulating cell growth and division. ERK1/2 regulates transcription factors and gene expression that promotes cell proliferation [71]. MEK phosphorylates ERK1/2 at Thr202/Tyr204 [24,72,73] (Appendix A and Table 1). Total levels of ERK1/2 were also significantly increased in DM1 human skeletal muscle biopsies (*p* = 0.02 in BB muscle biopsies; *p* = 0.01 in TA muscle biopsies) [45] and phosphorylated levels at Thr202/Tyr204 were increased in TA distal skeletal muscle biopsies (*p* = 0.03) [45] and human myoblasts (*p* < 0.01) [45] (Appendix A, Table 1). In addition, phosphorylated levels were positively correlated with CTG repeat length in a DM1 Lymphoblastoid B-Cell Line (LBCLs) [35] (Table 1). These results indicate that the MEK/ERK pathway may be upregulated in DM1.

This pathway was further evaluated upon insulin stimulation, where phosphorylated ERK1/2 (Thr202/Tyr204) levels remained unresponsive upon insulin signaling activation in DM1, while in control samples these levels significantly increased after stimulation [45,46] in human skeletal muscle biopsies and satellite cell-derived myotubes (Table 1).

#### 3.5.5. Myoblast Differentiation and Proliferation

DMPK is the DM1 central protein and evidence suggests that due to the occurrence of CUG-expanded mRNA that forms hairpin-like structures and accumulates in the nucleus, the DMPK protein levels are diminished. Indeed, all studies reported a significant decrease in the total levels of DMPK in DM1 compared to controls (Appendix A, Table 1) [24,28,30,50,57,58,59,62].

The myoblast proliferation and differentiation signaling alterations observed in DM1 are summarized in Appendix A [24,47,49]. Firstly, DMPK was decreased during the differentiation process in human fetus-derived myoblasts in DM1 compared to controls. The CDK4 protein levels were unchanged in DM1 but increased in control during differentiation (Appendix A and Table 1) [49,61]. CUGBP1, one of CDK4’s targets, had both total and Ser302 phosphorylated levels decrease during human myoblast differentiation (Appendix A and Table 1) [49]. Rb protein, another CDK4 downstream target, was shown to be slightly increased during the first two days of differentiation of DM1 fetus-derived myoblasts compared to controls. Rb, in the 4th to 6th day of myoblast differentiation, was completely dephosphorylated in DM1, similar to controls (Appendix A and Table 1) [24].

The AKT/mTOR pathway studies in DM1 showed that AKT has increased phosphorylated levels (Ser473) in proliferating DM1 myoblasts but decreased levels in DM1 differentiating myoblasts (Appendix A and Table 1) [49]. Additionally, S6K1 phosphorylated levels (Thr421/Ser424) also appeared mostly reduced during human myoblast differentiation in DM1 (*p* < 0.01) [24,47], except for myoblasts with 1800 CTG repeats [47] (Appendix A and Table 1). Phosphorylated CUGBP1 at Ser38, one of the downstream targets of AKT, appeared increased during human myoblast proliferation (Table 1) [49], and interactions with these two proteins were also increased during proliferation in DM1 [49]. Another downstream target of AKT, GSK3β, increased total levels during all steps of myoblast differentiation (Appendix A and Table 1) [33]. Concerning the MEK/ERK pathway, the p-ERK1/2 (Thr202) levels were increased in DM1 myoblasts through almost all of the differentiation process, but especially at the beginning of the differentiation (Appendix A and Table 1) [24]. Further, its effector MEK also presented increased phosphorylation levels (Ser218/Ser222) in the beginning of myoblast differentiation (Appendix A and Table 1) [24]; p38MAPK, however, presented reduced phosphorylated levels (Thr180/Tyr182) during the DM1 myoblast differentiation process (Appendix A and Table 1) [24].

#### 3.5.6. Other Relevant Signaling Alterations

The PKC kinase has been found to be related to MEK/ERK pathway activation [72] and CUGBP1 increased steady-state levels and phosphorylation [37]. Three isoforms of PKC were evaluated in DM1: PKCα, PKCβII, and PKCθ. In mice skeletal muscle biopsy, PKCα/βII and PKCθ total levels and PKCα/βII phosphorylated levels at Thr638/Thr641 were similar in DM1 and controls (Appendix A and Table 1) [36] but they were increased in mice and human cardiac muscle biopsies and in COS M6 cell lines with expanded CUG repeats (Table 1) [37]. Regarding PKCθ, phosphorylated levels at Thr538 were found significantly increased in DM1 compared to control mice skeletal muscle biopsies (*p* = 0.05) (Appendix A) [36].

Phospholamban (PLN) is another downstream target of DMPK. This protein regulates the sarcoplasmic reticulum calcium pump and, when phosphorylated, increases calcium reuptake [73]. DMPK phosphorylates PLN at Ser16 [39]. In DMPK knockout mice cardiac tissue biopsies, PLN phosphorylated levels at Ser16 were decreased before and after stimulation with the β-adrenergic agonist isoproterenol compared to controls (*p* < 0.05) (Table 1) [39].

One protein that is influenced by calcium levels is calcineurin, a Ca^2+^-activated serine/threonine phosphatase (PP2B) which is an important stimulator of muscle growth, hypertrophy, and remodeling [74,75]. As observed in DM1 HSALR mice skeletal muscle tissue biopsies, this phosphatase activity and total protein levels were significantly increased about two-fold compared to controls (Appendix A and Table 1) [43].

PKR and PKR-like endoplasmic reticulum kinase (PERK) proteins are kinases that respond upon stress signals. Both phosphorylate and inhibit the initiation factor 2 subunit 1 (eIF2α) at Ser51, leading to protein synthesis suppression and apoptosis induction [76]. In DM1, phosphorylated levels of PKR [31,41] and PERK [31] are increased in human skeletal muscle biopsies and myoblasts. Also, protein CDK6, a kinase that promotes eIF2α translation of transcription factors for cell cycle progression [77], is also increased in human myoblasts (Appendix A and Table 1) [23]. EIF2ɑ is involved in the initiation of eukaryotic protein synthesis and can be phosphorylated and inhibited at Ser51 by the previously described PKR and PERK proteins, leading to autophagy and suppression of protein synthesis [76]. In DM1, phosphorylated levels of eIF2ɑ at Ser51 are increased in human skeletal muscle biopsies (Appendix A and Table 1), human myoblasts (Appendix A and Table 1), and CHO cell lines (Table 1) [31]. NF-kB, a family of inducible transcription factors that mediate pro-inflammatory gene induction [78], is indirectly activated by PKR, through the phosphorylation of IkBα (a NF-kB inhibitor) at Ser32, which targets the protein for degradation [78]. In mice C2C12 myoblasts with 800 CUGexp RNA, NF-kB has similar total levels when compared to control and increased phosphorylated levels at Ser536 (*p* < 0.01) (Table 1) [41]. IkBα has increased phosphorylated levels at Ser32 (*p* < 0.01) [42], which results in increased protein degradation, being consistent with the significant decrease observed in its total levels (*p* < 0.01) (Table 1) [41].

Several proteins involved in cell proliferation are altered in DM1. PKM2 is a pyruvate kinase isomer that catalyzes the final reaction of glycolysis, usually expressed in tissues such as tumor cells, embryonic tissues, and testis. Its activity promotes cancer cell proliferation and growth [79]. In DM1, the PKM2 is aberrantly expressed in muscle, with increased total levels in C2C12 myoblasts and human skeletal muscle biopsies (Appendix A and Table 1). Signal transducer and activator of transcription 3 (Stat3) is a transducer of cytokines and growth factors. When phosphorylated, Stat3 is active and regulates target genes related to cell growth, proliferation, differentiation, and apoptosis [80]. Aberrant phosphorylation of Stat3 has been related to malignant cell transformation induction [80]. In DM1, total and phosphorylated Stat3 levels at Ser727 were found increased in mice skeletal muscle biopsies (Appendix A and Table 1) [40]. Lastly, the Rb protein is a tumor suppressor that negatively regulates the G1/S cell cycle transition [81]. It becomes inhibited when phosphorylated, enabling cell cycle progression. Rb phosphorylated levels were increased in DM1 human myoblasts at an unknown residue (Appendix A and Table 1) [24] but decreased in neural stem cells at Ser801/Ser811 (*p* < 0.05) (Table 1) [27].

### 3.6. Discussion

In this systematic review, we gathered evidence suggesting deregulation of several protein kinases, protein phosphatases, and phosphoproteins deregulated in DM1, as well several signaling pathways that might be impaired, contributing to the clinical manifestations and symptoms commonly observed in patients with DM1. Some heterogeneous and conflicting results were reported depending on the model system used. In fact, several in vivo and in vitro biological models have been used to study the molecular mechanisms underlying DM1, with several advantages/disadvantages. For instance, transgenic mice are a biological model of choice that recreates key molecular and phenotypic features of DM1 [82,83]. However, some of the limitations for the use of mouse models are that the levels of repeat instability are decreased in comparison to what is observed in germ lines and somatic tissues in human cell models, possibly due to the use of a relatively shorter number of repeats than observed in humans, especially in more severe phenotypes [84]. Human-derived cell models are of great utility for the study of DM1, since they can express the whole range of repeat lengths while staying at their genomic context and reproducing the characteristic features of DM1 [85,86,87]. One of the major limitations of primary cells, in general, is the limited number of times they can divide until they enter a state of replicative senescence. In DM1, this effect is even more accentuated since cells may enter a state of early senescence compared to the control cells. At the same time, primary cells have great variability between individuals, which alters the growth conditions of the cell line considering age, the tissue of origin, and the degree of dysfunction [88,89,90]. Therefore, to increase the robustness of the data and integrate the research findings related to DM1, both the use of mouse models and human cell lines should be considered. Further, also reported were variations in the results obtained in studies performed using human muscle samples, which may be related to the sample collection (i.e., biopsies, post-mortem). Moreover, differences between the type of muscle group—for instance, between BB and TA—are also observed. These differences are, in fact, expected in DM1, given that this muscular dystrophy is characterized by progressive and symmetrical distal muscle weakness [5].

Due to the high variability in methodologies and DM1 models used, a meta-analysis was not possible to perform. The results of this review clearly demonstrate that several signaling pathways are impaired in DM1, such as the AKT/mTOR, MEK/ERK, PKC, and AMPK signaling pathways. These pathways are responsible for regulating different key cellular mechanisms, such as cell cycle, cell proliferation and differentiation, apoptosis, autophagy, glucose metabolism, and stress response [70,71,91].

DM1 is characterized by genetic alterations in the *DMPK* gene, specifically at the 3′UTR region, and patients with DM1 have abnormally expressed CTG repeats. This gene encodes for the DMPK protein, which was found decreased in patients with DM1 in all studies included in this review [24,28,30,50]. One plausible explanation is that DMPK mRNA transcripts with expanded CUG tend to be retained at the nucleus as nuclear aggregates, called foci, preventing DMPK mRNA translocation to the cytoplasm, leading to decreased DMPK expression and, consequently, loss of function [4]. Also, around a 50% decrease was reported in DMPK mRNA levels in patients with DM1 [58,92]. Targets of DMPK such as PLN also have decreased phosphorylated levels in DM1 mice heart biopsies compared to controls [39], which can explain the increased probability for patients with DM1 to develop cardiac conduction abnormalities. Surprisingly, to date, there have been no results to determine the functional relation between the DMPK protein with its target CUGBP1 [19], despite being two of the most altered proteins in this disorder. These studies are very important and should be addressed in the near future. Currently, there is no reported evidence suggesting the direct role of DMPK in other kinases and phosphatases highlighted in this manuscript. Although DMPK might not be impacting directly other kinases and phosphatases, it is expected that by altering several signaling pathways, it impacts indirectly their expression and activity, which might be underlying some of DM1’s heterogenous symptoms.

MBNL1 and CUGBP1 are two RNA-binding proteins that were found deregulated in DM1, contributing to abnormal alternative splicing of several different pre-mRNAs [69]. Studies that addressed CUGBP1 in DM1 detected increased steady-state levels and hyperphosphorylation that can be attributed to an increase in PKC expression and activity [37]. However, a study that used PKCα/β knockout mice skeletal muscle biopsies presenting RNA toxicity did not observe a difference in the skeletal muscle phenotypes compared to a wildtype, suggesting that the phenotypes are independent of these proteins [36]. Another PKC isoform named PKCθ, predominantly expressed in skeletal muscle, demonstrated increased phosphorylated levels in skeletal muscle biopsies of DM1 patients. PKCθ is suggested to be involved in biological events associated with DM1 pathology, such as myoblast differentiation and chloride channel function modulation [93,94,95]. Although the mechanisms underlying PKC upregulation mediated by the CUG expanded repeats remain to be elucidated, PKCθ may be an important target to explore in DM1. Additionally, the relationship between PKCθ phosphorylation and CUGBP1 increased phosphorylation should be explored.

CUGBP1 is also regulated by AKT and cyclinD3/CDK4/6 phosphorylation, altering its affinity for certain mRNAs. As previously described, in DM1 human myoblasts, CUGBP1 has decreased interaction with cyclin D3 in myotubes, leading to increased formation of inhibited translational complexes with phosphorylated eIF2α (inhibited) [31]. These complexes aggregate into stress granules in the cytoplasm, trapping mRNAs that code for proteins involved in DNA damage repair and also the remodeling factor MRG15, contributing to the progressive muscle loss in patients with DM1 [31]. Additionally, the formation of stress granules with RNA binding proteins has been reported in neurodegenerative disorders such as Amyotrophic Lateral Sclerosis and Spinal Muscular Atrophy [96].

The AKT/mTOR pathway was reported to be impaired in DM1, especially under stimulation with insulin. Insulin insensitivity and resistance is a common endocrine abnormality observed in patients with DM1 [97]. AKT and mTOR, which are important stimulators of anabolic pathways such as glucose uptake, glycogen storage, and protein synthesis, did not respond to insulin stimulation and, therefore, were found decreased in DM1. FOXO1, AS160, and GSK3α/β, which should be inhibited upon insulin stimulation, presented increased activity in DM1, leading to increased apoptosis, oxidative stress, and protein degradation. In Figure 4, we propose a mechanism by which the insulin pathway is impaired in DM1, considering the results gathered within this review. In addition, the MEK/ERK pathway is responsible for the growth-promoting effects of insulin [98]. Together they regulate several biological processes, such as transcription, protein synthesis, cell growth, and differentiation [99]. ERK1/2 was also found decreased in DM1 upon insulin stimulation [46] (Figure 4).

Studies on alternative splicing detected abnormal splicing of the InsR mRNA, where patients with DM1 have increased expression of the fetal isoforms instead of the adult isoforms, which can partially explain the decreased response upon insulin stimulation [46,100]. Another cause could be the increase in basal phosphorylation levels of AKT, ERK1/2 and GSK3β without insulin stimulation, as observed in distal skeletal muscle biopsy from patients with DM1, which could impair further stimulation upon insulin signaling. Lack of insulin stimulation has been linked to loss of muscle strength and muscle mass [101], symptoms typically associated with progressive muscle wasting observed in DM1. Together, the dysfunction of the insulin pathway in DM1 may explain why these patients are more likely to develop metabolic complications such as insulin resistance [100,102].

One of the insulin pathway antagonists is the AMPK signaling pathway. This pathway is one of the major regulators of cell metabolism in eukaryotes and is tightly related by autophagy [70]. DM1 HSALR mice skeletal muscle biopsies did not respond upon starvation conditions [26]. In Figure 5, we propose a mechanism by which the AMPK pathway is impaired in DM1 considering the results gathered in this review. While the upstream effectors of AMPK, LKB1, and TAK1 did not present statistically significant differences between DM1 and controls, total and phosphorylated levels of CaMKIIβM, also an upstream effector of AMPK, which is activated in response of Ca^2+^ flux [70], were significantly decreased in DM1. AMPK and its downstream target ULK1 also appeared with decreased phosphorylated active levels, which may be related to CaMKIIβM decreased activity (Figure 5B).

Previous studies on alternative splicing in DM1 detected abnormalities in the splicing of CaMKII mRNA [103], which could explain the decreased response. However, it is unknown whether these results are due to an impaired response induced by starvation or lack of Ca^2+^ flux since the latter was not reported. Further, the downstream target of mTOR, S6K1 as well as its target RpS6, presented increased phosphorylated levels upon starvation, while mTOR levels were similar between DM1 and controls [26]. This could be either due to a mechanism independent of mTOR or to increased activity and interaction of mTOR and S6K1, which was not ascertained in the study [26]. Together, these results suggest autophagy and apoptosis perturbations and an aberrant increase of anabolic pathways, increased protein synthesis and cell growth [26,70], which can lead to muscle alterations such as muscle atrophy and myotonia. In fact, DM1 mice treated with the AMPK agonist 5-aminoimidazole-4-carboxamide ribonucleoside (AICAR) and the mTORC1 inhibitor rapamycin normalized muscle weakness and reduced myotonia [26].

The regulation of skeletal muscle formation (myogenesis) involves several different signaling pathways that are tightly regulated, controlling cell differentiation and proliferation [104,105,106]. In DM1, pathways that promote differentiation are decreased, while pathways that promote proliferation are increased during differentiation, which significantly impairs and delays myogenesis [107]. Previous studies using C2C12 cells demonstrated that DMPK has a role in cell morphology regulation during myogenesis and is necessary for myogenin expression during differentiation of myoblasts to myotubes [108]. DMPK is markedly decreased during differentiation in DM1, and mRNA studies corroborate that the decreased levels of DMPK mRNA transcripts lead to a delay in DM1 myoblast maturation due to nuclear retention of the transcripts [58]. CUGBP1 increased phosphorylation by cyclin D3/CDK4/CDK6 enables the translation of mRNAs important for myoblast differentiation [106,109]. However, in DM1, CUGBP1 is decreased and interacts less with cyclin D3 due to decreased levels of cyclin D3 and CDK4 during differentiation [49]. Moreover, an increased interaction between CUGBP1 and AKT is found in DM1 myoblasts [49], increasing the levels of proteins that are strong stimulators of cell proliferation such as cyclin D1 [109]. Tumor suppressors such as Rb protein are important for cell cycle and differentiation. CDK4 kinase phosphorylates and inhibits the Rb protein, enabling the cell cycle progression [110,111]. In DM1, there was an increase in the phosphorylated inactive form of Rb during differentiation, enabling cell proliferation, a feature observed in different types of cancer [112]. The MEK/ERK pathway mainly promotes and regulates cellular proliferation [105], and in DM1 is abnormally active in the early stages of DM1 myoblast differentiation [24]. Additionally, the aberrant expression of other cell proliferation stimulators, such as PKR, PERK, and PKM2, observed in DM1 are also observed in cancer. This may explain why patients with DM1 present increased cancer susceptibility [113,114,115].

In this present systematic review, one of the limitations observed in the studies included is the lack of reporting on statistical significance to assess the differences between DM1 models and controls, which occurred only in 10 out of 41 studies included. Although this does not take credibility to the results, p-level reporting could increase the robustness of the data for this systematic review. It is also worth noting the lack of research on some main phosphatases that can participate in the regulation of the different signaling pathways that are dysfunctional in DM1. However, calcineurin presented robust results, with significantly increased levels in DM1 mice models. Surprisingly, this observation was considered a beneficial compensatory mechanism in DM1, since the normalization of calcineurin levels resulted in aggravated phenotypes of abnormal alternative splicing [43]. Calcineurin overexpression has also been reported to ameliorate the dystrophic phenotype in Duchenne Muscular Dystrophy (MIM # 310200) mice models [116].

In order to increase the knowledge of the role of protein phosphatases in the DM1 underlying molecular mechanisms, the study of other phosphatases should be considered in the future, such as protein phosphatase 1 and 2 (PP1/PP2A), which account for more than 90% of the protein phosphatase activity in eukaryotic cells [12]. They may reduce or increase the activity of certain proteins that have been shown to be altered in DM1, such as AKT, mTOR, or GSK3, which in turn influences the activity of their downstream targets [117,118]. Also, PP1 and its regulatory proteins have been found to be important participants in nuclear events, such as cell cycle and transcription regulation [12], which has been reported to be dysfunctional in DM1 in this review. Other phosphatases, such as Phosphatase and Tensin Homolog (PTEN), which indirectly dephosphorylates AKT, have been shown to be increased in models of muscle atrophy [119]. However, there is a lack of evidence of how these proteins interact and how their protein levels and activity vary in DM1 human or cellular models. Altogether, the described phosphatases may be good candidates for future research to unravel new molecular mechanisms and signaling pathways altered in DM1.

## 4. Conclusions

In conclusion, this review provided a compilation of the altered protein phosphorylation events in several different kinases, phosphatases, and phosphoproteins that participate in signaling pathways, which apparently are deregulated in DM1. In Figure 6, we compile the major protein phosphorylation alterations in DM1 reported in the studies included in this review and the corresponding pathomechanisms and clinical observations.

Key cellular pathways such as AKT/mTOR and AMPK were underactive in DM1, especially upon exposure of stimulating factors such as insulin/IGF-1 or decreased ATP levels, leading to glucose metabolism impairments and decreased autophagy upon starvation. Also, several pathways and proteins, such as MEK/ERK, NF-kB, PKR, and PERK, were overexpressed and overactive, contributing to increased proliferation and delayed myogenesis in DM1. These findings explain the heterogeneity of symptoms and muscular and extra muscular clinical manifestations in DM1, such as muscle weakness, delayed differentiation, insulin resistance, and cardiac conduction problems. Also, the findings provide useful insights into the different phosphorylation abnormalities in regulation of several pathways, which can be good candidates for the development of innovative therapies in future clinical trials.

## Figures and Tables

**Figure 1 ijms-24-03091-f001:**
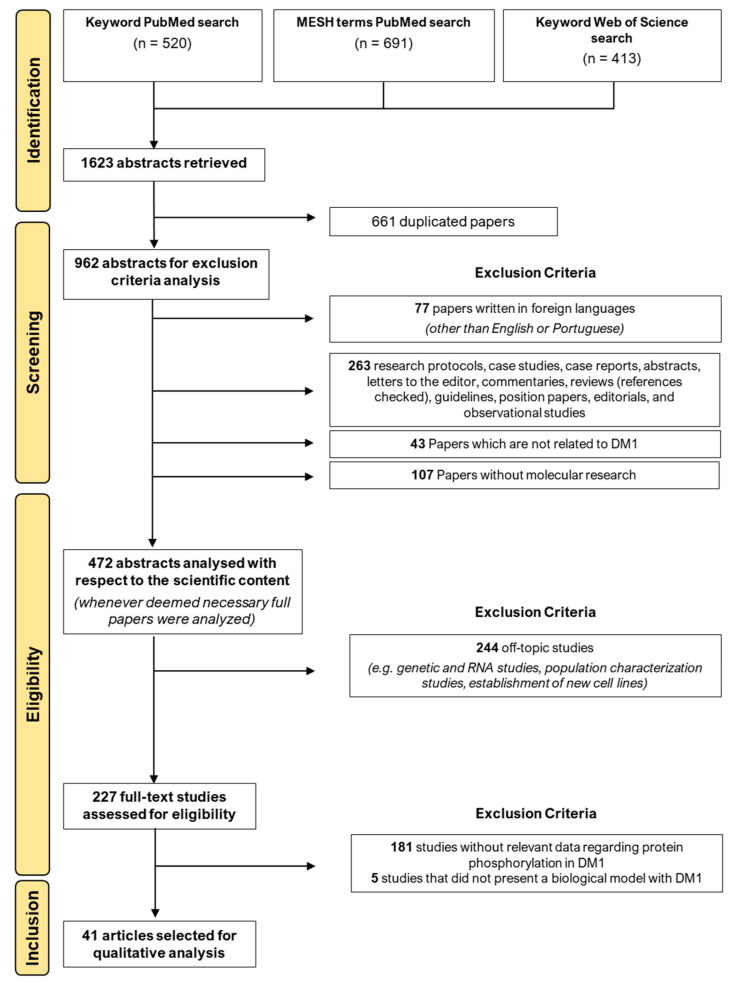
PRISMA flow diagram from the results of the systematic search strategy and study selection.

**Figure 2 ijms-24-03091-f002:**
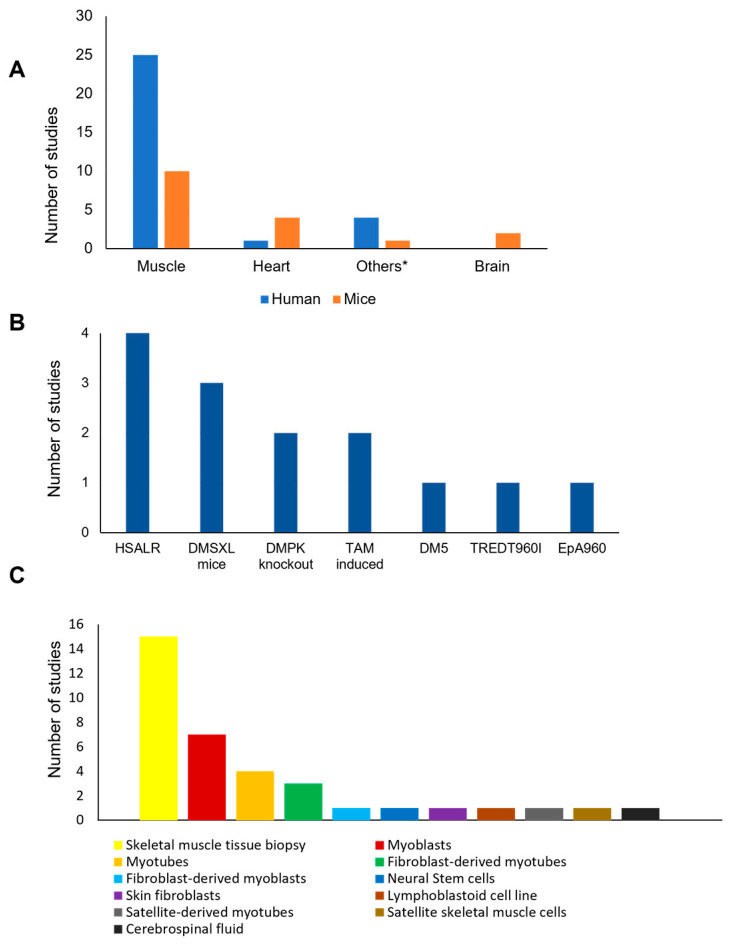
Type of biological sample used and diversity of mice models, human cell types, and body fluids used in the selected studies. (**A**) Type of organs and tissues used across the studies; (**B**) DM1 mice models used in the selected studies; (**C**) Type of human samples used across the studies. DM5—RNA toxicity model; DMPK knockout—Mice models without the dystrophia myotonica protein kinase gene; DMSXL—Transgenic mice carrying >1000 CTG repeats; HSALR—Transgenic mice expressing the human skeletal actin gene with long CUG repeat length; TAM induced—Tamoxifen-inducible and heart-specific EpA960(R) RNA expression model; TREDT960I—Tetracycline-inducible transgene model; EpA960—interrupted 960-CTG expansion model. * Other organ and tissue derived samples.

**Figure 3 ijms-24-03091-f003:**
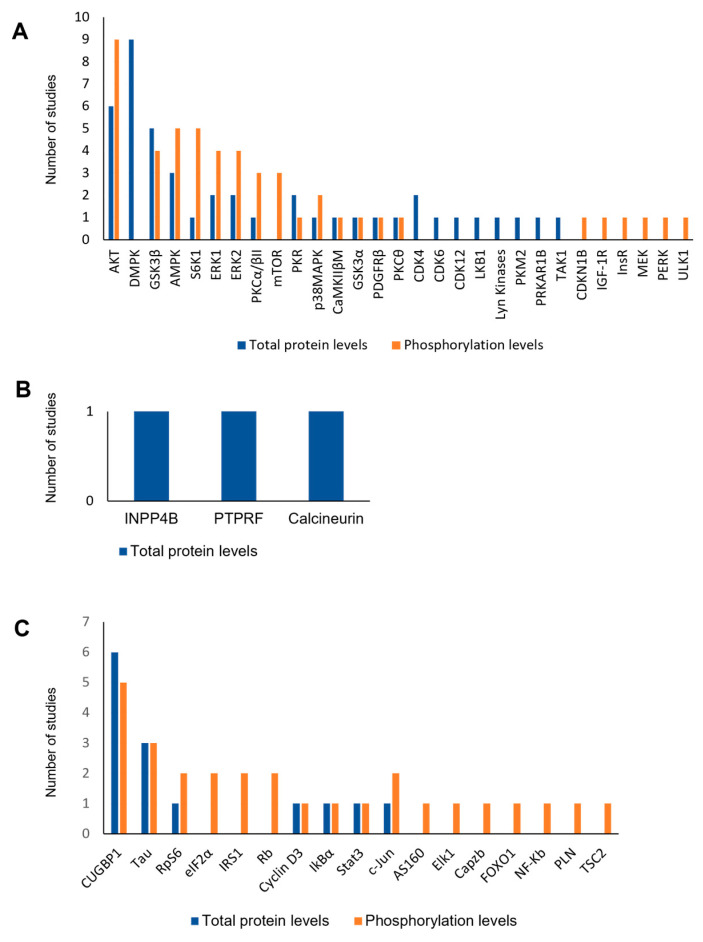
Distribution of protein kinases, protein phosphatases, and phosphoproteins evaluated across the studies of this systematic review. Both total levels and phosphorylation levels were evaluated. (**A**) Kinases; (**B**) Phosphatases; (**C**) Phosphoproteins. Abbreviations: AKT—Protein kinase B; AMPK—5′ AMP-activated protein kinase; AS160—TBC1 domain family member 4; CaMKIIβM—Ca2+/calmodulin dependent protein kinase; CDKN1B—Cyclin-dependent kinase inhibitor 1B; CDK4/6/12—Cyclin-dependent kinase 4/6/12; CUGBP1—CUG triplet repeat RNA binding protein 1; DMPK—Dystrophia myotonica protein kinase; eIF2α—Eukaryotic translation initiation factor 2 subunit 1; ERK 1/2—Extracellular signal-regulated kinases 1/2; FOXO1—Forkhead box protein O1; GSK3—Glycogen synthase kinase 3; IGF-1R—Insulin-like growth factor 1 receptor; IkBα—Nuclear factor of kappa light polypeptide gene enhancer in B-cells inhibitor, alpha; INPP4B—Type II inositol 3,4-bisphosphate 4-phosphatase; InsR—Insulin receptor; IRS1—Insulin receptor substrate 1; LKB1—Liver kinase B1; MEK—Mitogen-activated protein kinase kinase; mTOR—Mammalian target of rapamycin; NF-kB—Nuclear factor kappa-light-chain-enhancer of activated B cells; PDGFRβ—Platelet-derived growth factor receptor beta; PERK—Protein kinase R (PKR)-like endoplasmic reticulum kinase; PKC—Protein kinase C; PKM2—Pyruvate kinase isozyme M2; PKR—Protein kinase; PLN—Phospholamban; PRKAR1B—cAMP-dependent protein kinase type I-beta regulatory subunit; PTPRF—Receptor-type tyrosine-protein phosphatase F; p38MAPK—p38 Mitogen-activated protein kinase; Rb—Retinoblastoma protein; RpS6—Ribosomal protein S6; Stat3—Signal transducer and activator of transcription 3; S6K1—Ribosomal protein S6 kinase beta-1; TAK1—Mitogen-activated protein kinase kinase kinase 7; TSC2—Tuberous Sclerosis Complex 2; ULK1—Unc-51-like autophagy activating kinase.

**Figure 4 ijms-24-03091-f004:**
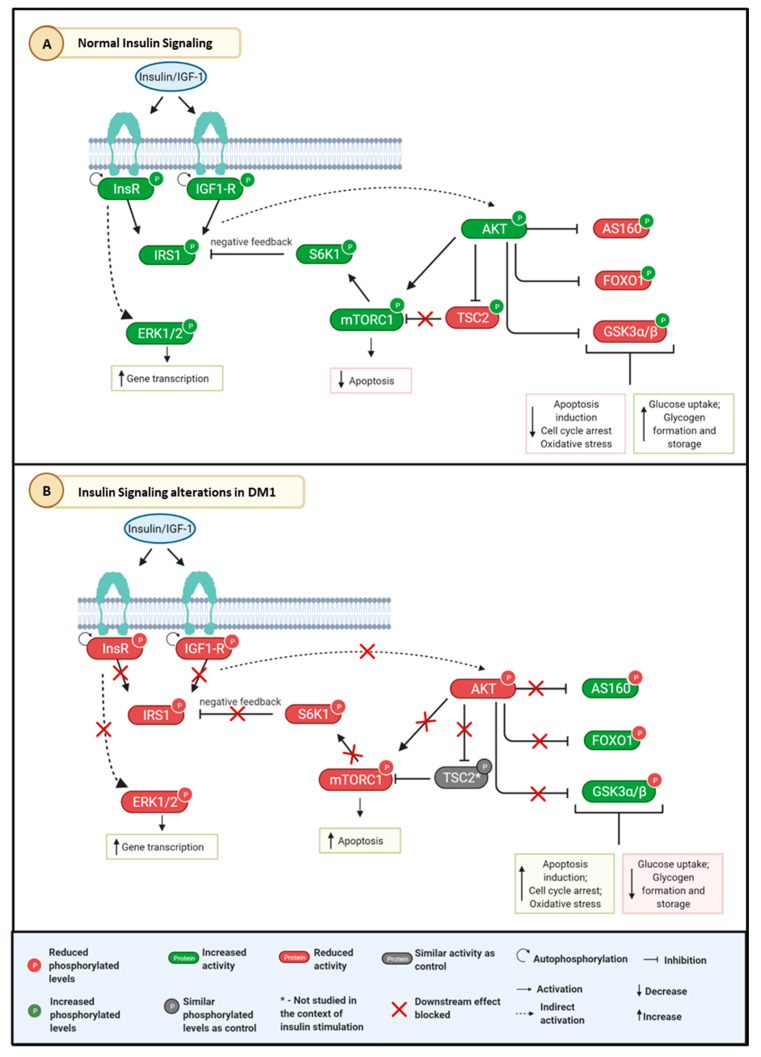
Schematic representation of the proposed mechanism of insulin signaling dysfunction in DM1. (**A**) Normal insulin signaling pathway; (**B**) Insulin signaling alterations in DM1 according to the results. The AKT/mTOR and MEK/ERK pathway have decreased levels of phosphorylation and activity, contributing to decreased glucose uptake and gene transcription, as well as increased apoptosis. Created using Biorender.com. Abbreviations: AKT—Protein kinase B; AS160—TBC1 domain family member 4; ERK 1/2—Extracellular signal-regulated kinases 1/2; FOXO1—Forkhead box protein O1; GSK3—Glycogen synthase kinase 3; IGF-1R—Insulin-like growth factor 1 receptor; InsR—Insulin receptor; IRS1—Insulin receptor substrate 1; mTOR—Mammalian target of rapamycin; S6K1—Ribosomal protein S6 kinase beta-1; TSC2—Tuberous Sclerosis Complex 2.

**Figure 5 ijms-24-03091-f005:**
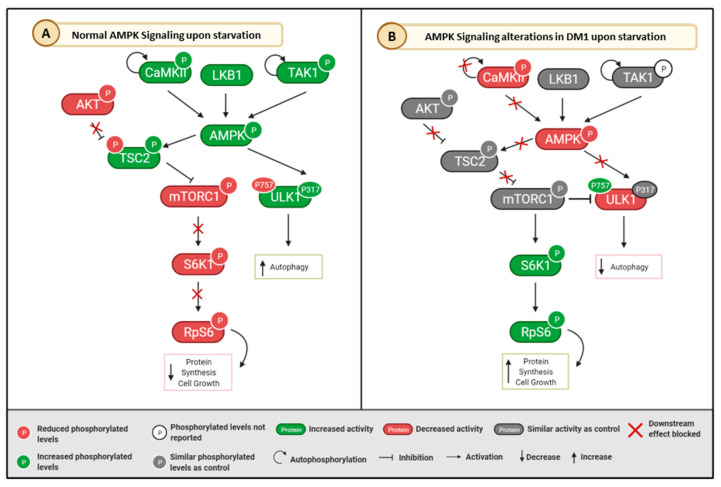
Schematic representation of the proposed mechanism of AMPK signaling dysfunction in DM1. (**A**) Normal AMPK signaling pathway; (**B**) AMPK Signaling alterations in DM1 upon starvation, according to the results. In DM1, an abnormal activation of mTORC1 pathway and inhibition of the AMPK pathway during ATP decrease is observed. Created using Biorender.com. Abbreviations: AKT—Protein kinase B; AMPK—5′ AMP-activated protein kinase; CaMKII—Ca^2+^/calmodulin dependent protein kinase; LKB1—Liver kinase B1; mTOR—Mammalian target of rapamycin; RpS6—Ribosomal protein S6; S6K1—Ribosomal protein S6 kinase beta-1; TAK1—Mitogen-activated protein kinase kinase kinase 7; TSC2—Tuberous Sclerosis Complex 2; ULK1—Unc-51-like autophagy activating kinase; enhancer in B-cells inhibitor, alpha.

**Figure 6 ijms-24-03091-f006:**
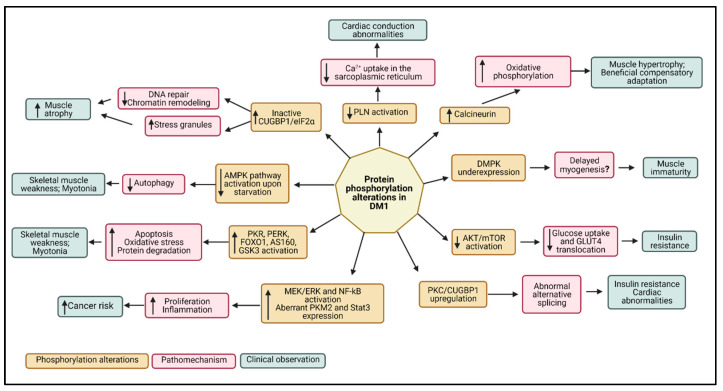
Schematic representation of the major protein phosphorylation and molecular alterations in DM1, corresponding pathomechanisms, and clinical manifestations. Created using Biorender.com. Symbols: ↑, increased; ↓, decreased.

**Table 1 ijms-24-03091-t001:** Summary of the main findings of the different proteins’ total and phosphorylation levels in DM1.

Protein	Biological Model	Organ/Tissue/Cell Line	Total Levels (DM1 vs. CTL)	Phosphorylated Levels (DM1 vs. CTL)	Phosphorylated Residue	Protein Activity (DM1 vs. CTL)		
**Protein Kinases**								
AKT [26,27,30,35,38,45,46,47,48,49]	Mice	Muscle tissue	NR	Similar in fed mice [26] Similar in mice with 24 h starvation conditions [26]	Ser473 Ser473			
		Heart tissue	NR	Decreased upon insulin pathway activation [38]	Ser473			
	Human	Satellite-cell derived myotubes	NR	Decreased upon insulin pathway activation [45]	Thr308			
		Muscle tissue	Similar [26]; Similar in BB muscle [45]; Increased in TA muscle [45]	Similar in BB muscle [45] Increased in TA muscle [45] Decreased upon insulin pathway activation [45]	Thr308 Thr308 Thr308			
				Similar [26] Decreased upon insulin pathway activation [45]	Ser473 Ser473			
		Myoblasts	Decreased in cells with 3200 repeats [47]; Similar in cells with 1800 repeats [47]; Similar [49]	Increased [49]	Ser473	Increased interaction with CUGBP1 [49]		
		Myoblasts to myotubes	Similar in differentiation [49]	Decreased in differentiation [49]	Ser473			
		Fibroblast-derived myotubes	NR	Similar levels upon starvation conditions [26] Decreased [48]	Ser473 Ser473			
		Neural stem cells	Similar [27]	Similar [27]	Ser473			
		Fibroblasts	NR	Decreased [30]	Ser473			
		Lymphoblastoid cell lines	Similar across different CTG repeats * [35]	Similar across different CTG repeats * [35]	Ser473			
		Myotubes	Similar [49]	Decreased [49]	Ser473			
AMPK [26,27,40,44,48]	Mice	Muscle tissue	Similar in mice with moderate muscle wasting [40]; Increased in mice with severe muscle wasting [40]; Similar [44]	Similar in 45 h starvation conditions [26] Decreased in 24 h starvation conditions [26] Similar in fed mice [26] Increased in mice with severe muscle wasting [40] Similar in mice with moderate muscle wasting [40] Decreased [44]	Thr172 Thr172 Thr172 Thr172 Thr172 Thr172			
	Human	Muscle tissue	Similar [26,40]	Similar [26,40]	Thr172			
		Fibroblast-derived myoblasts	Similar [44]	Decreased [44]	Thr172			
		Fibroblast-derived myotubes	NR	Similar levels upon starvation conditions [26] Similar [48]	Thr172 Thr172			
		Neural stem cells	Similar [27]	Similar [27]	Thr172			
CaMKIIβM [26]	Mice	Muscle tissue	Decreased in fed mice [26]; Decreased in mice with 24 h starvation conditions [26]	Decreased in fed mice [26] Decreased in mice with 24 h starvation conditions [26]	Thr286 Thr286			
CDKN1B [40]	Mice	Muscle tissue	NR	Increased [40]	Thr198			
CDK4 [49]	Human	Myoblasts to myotubes	Decreased in differentiation [49] Similar in differentiation [61]	NR NR	NR NR	Increased activity in differentiation [61]		
CDK6 [23]	Human	Myoblasts	Increased [23]	NR	NR			
CDK12 [34]	Human	Muscle tissue	Increased [34]	NR	NR			
DMPK [24,28,30,50,57,58,59,62]	Human COS 7 cells	Myoblasts to myotubes	Decreased in differentiation [24]	NR	NR			
		Myotubes	Decreased [28]	NR	NR			
		Muscle tissue	Decreased [28,50,57,59,62]	NR	NR			
		Myoblasts	Decreased [58]	NR	NR			
		Fibroblasts Cell line	Decreased [30] Decreased [56]	NR NR	NR NR			
ERK1/2 [35,45,46]	Human	Lymphoblastoid cell lines	Similar across different CTG repeat lengths * [35]	Increased across different CTG repeats * [35]	Thr202/Tyr204			
		Muscle tissue	Increased in BB and TA muscle [45]	Similar in BB muscle [45] Increased in TA muscle [45] Decreased upon insulin pathway activation [46]	Thr202/Tyr204 Thr202/Tyr204 Thr202/Tyr204			
ERK1 [24,45]	Human	Myoblasts	NR	Increased [24]	Tyr204			
		Myoblasts to myotubes	NR	Increased during differentiation [24]	Tyr204			
		Satellite-cell derived myotubes	NR	Similar upon insulin pathway activation [45]	Thr202			
ERK2 [24,45]	Human	Myoblasts	NR	Increased [24]	NR			
		Myoblasts to myotubes	NR	Increased at first days of differentiation [24]	NR			
		Satellite-cell derived myotubes	NR	Decreased upon insulin pathway activation [45]	Tyr204			
GSK3α [27]	Human	Neural stem cells	Similar [27]	Decreased [27]	Ser21			
GSK3β [27,33,36,38,45,53]	Mice	Heart tissue	NR	Decreased upon insulin pathway activation [38]	Ser9			
		Muscle tissue	Increased in 1-month-old mice [33,36]; Increased in 6-month-old mice [33]; Increased [53]	NR	NR			
	Human	Neural stem cells	Similar [27]	Decreased [27]	Ser9			
		Muscle tissue	Increased [33]; Similar in BB muscle [45]; Decreased in TA muscle [45]	Decreased [33]	Ser9			
				Increased [33] Similar in BB muscle [45] Increased in TA muscle [45]	Tyr216 Tyr216 Tyr216			
		Myoblasts	Increased [53]	NR	NR			
		Satellite-cell derived myotubes	NR	Similar upon insulin pathway activation [45]	Tyr216			
		Myoblasts to myotubes	Increased during differentiation [33]	NR	NR			
IGF-1R [38]	Mice	Muscle tissue	NR	Decreased upon insulin pathway activation [38]	Tyr1135/Tyr1136			
		Heart tissue	NR	Decreased upon insulin pathway activation [38]	Tyr1135/Tyr1136			
InsR [38]	Mice	Muscle tissue	NR	Decreased upon insulin pathway activation [38]	Tyr1150/Tyr1151			
		Heart tissue	NR	Decreased upon insulin pathway activation [38]	Tyr1150/Tyr1151			
LKB1 [26]	Mice	Muscle tissue	Similar in fed and 24 h starvation conditions [26]	NR	NR			
Lyn Kinases [25]	Human	Myotubes	Increased [25]	NR	NR	Increased nuclear activity and Tyr phosphorylation [25]		
MEK [24]	Human	Myoblasts	NR	Increased [24]	Ser218/Ser222			
		Myoblasts to myotubes	NR	Increased at first days of differentiation [24]	Ser218/Ser222			
mTOR [26,46,51]	Mice	Muscle tissue	NR	Similar in fed mice [26] Similar in mice in starvation conditions [26]	Ser2448 Ser2448			
	Human	Muscle tissue	NR	Decreased upon insulin pathway activation [46]	Ser2448			
		Satellite skeletal muscle cells	NR	Decreased [51]	Ser2448			
	IPSC-derived cells	Satellite skeletal muscle cells	NR	Decreased [51]	Ser2448			
PDGFRβ [40]	Mice	Muscle tissue	Increased in mice with severe muscle wasting [40] Similar in mice with severe muscle wasting [40]	Increased in mice with severe muscle wasting [40] Similar in mice with severe muscle wasting [40]	Tyr751 Tyr751			
	Human	Muscle tissue	Increased [40]	Increased [40]	Tyr751			
PERK [32]	Human	Muscle tissue	NR	Increased [32]	NR			
PKCα/βII [36,37,52]	Mice	Muscle tissue	Similar (PKCα) [36]	Similar [36]	Thr638/Thr641			
		Heart tissue	NR	Increased [52]	Thr638/Thr641			
	Human	Heart tissue	NR	Increased [37]	NR			
	COS M6 cells	Cell line	NR	Increased [37]	NR			
PKCθ [36]	Mice	Muscle tissue	Similar [36]	Increased [36]	Thr538			
PKM2 [29]	Mice	C2C12 myoblasts	Increased [29]	NR	NR			
	Human	Muscle tissue	Increased [29]	NR	NR			
PKR [31,41]	Mice	C2C12 myoblasts	Similar [41]	Increased [41]	NR			
	Human	Myoblasts	Increased [31]	NR	NR			
PRKAR1B [23]	Human	Myoblasts	Decreased [23]	NR	NR			
p38MAPK [24,30]	Human	Myoblasts	NR	Decreased [24]	Thr180/Tyr182			
		Myoblasts to myotubes	Similar in differentiation [24]	Decreased during differentiation [24]	Thr180/Tyr182			
		Fibroblasts	NR	Increased [30]	Thr180/Tyr182			
S6K1 [24,26,45,46,47]	Mice	Muscle tissue	NR	Similar in fed mice [26] Increased in 24 h starvation conditions [26]	Thr389 Thr389			
	Human	Muscle tissue	Similar in BB and TA muscle [45]	Increased [26]	Thr389			
				Similar in BB and TA muscle [45] Decreased upon insulin pathway activation [46]	Thr421/Ser424 Thr421/Ser424			
		Satellite-cell derived myotubes	NR	Similar upon insulin pathway activation [45]	Thr421/Ser424			
		Myoblasts	NR	Decreased [24] Decreased in cells with 3200 repeats [47] Increased in cells with 1800 repeats [47]	Thr421/Ser424 Thr421/Ser424 Thr421/Ser424			
		Myoblasts to myotubes	NR	Decreased during differentiation [24] Decreased during differentiation of cells with 3200 repeats [47] Increased during differentiation of cells with 1800 repeats [47]	Thr421/Ser424 Thr421/Ser424 Thr421/Ser424			
TAK1 [26]	Mice	Muscle tissue	Similar in fed and 24 h starvation conditions [26]	NR	NR			
ULK1 [26]	Mice	Muscle tissue	NR	Similar [26]	Ser757			
			NR	Similar [26]	Ser317			
**Protein Phosphatases**								
Calcineurin [43]	Mice	Muscle tissue	Increased [43]	NR	NR	NR		
INPP4B [23]	Human	Myoblasts	Increased [23]	NR	NR	NR		
PTPRF [23]	Human	Myoblasts	Increased [23]	NR	NR	NR		
**Phosphoproteins**								
AS160 [46]	Human	Muscle tissue	NR	Decreased upon insulin pathway activation [46]	Thr642			
c-Jun [40,60]	Mice	Muscle tissue C2C12 myoblasts	NR Similar upon MeHg-induced cytotoxicity [60]	Decreased [40] Increased upon MeHg-induced cytotoxicity [60]	Ser63 Ser63			
CUGBP1 [31,33,36,37,49,52,53]	Mice	Heart tissue	Increased [37,52]	Increased [37,52]	NR			
		Brain tissue	NR	Decreased [53]	Ser302			
		Muscle tissue	Increased [33,36]	NR	NR			
	Human	Myoblasts	Increased [31]	Decreased [31]	Ser302			
				Increased [49]	Ser28	Increased interaction with cyclin D1 mRNA [49]		
		Myoblasts to myotubes	Decreased in differentiation [49]	Decreased in differentiation [49]	Ser302	Decreased formation of active CUGBP1–eIF2 complex [49]		
		Myotubes	Decreased [49]	Decreased [49]	Ser302	Decreased interaction with cyclin D3 in myotubes [49]		
		Muscle tissue	Increased [33]	NR	NR			
		Fibroblast-derived myotubes	NR	Increased [37]	NR			
		Heart tissue	NR	Increased [37]	NR			
	CHO cell line	Cell line	Increased [31]	NR	NR			
Cyclin D3 [33]	Mice	Muscle tissue	Decreased [33]	NR	NR			
	Human	Muscle tissue	Decreased [33]	Increased [33]	Thr283			
eIF2α [31,32]	Human	Myoblasts	NR	Increased [31]	Ser51			
		Muscle tissue	NR	Increased [32]	Ser51			
	CHO cell line	Cell line	NR	Increased [31]	Ser51			
FOXO1 [46]	Human	Muscle tissue	NR	Decreased upon insulin pathway activation [46]	Thr24			
IkBα [41]	Mice	C2C12 myoblasts	Decreased [41]	Increased [41]	Ser32			
IRS1 [45,46]	Human	Muscle tissue	NR	Decreased upon insulin pathway activation [46]	Tyr612			
		Satellite-cell derived myotubes	NR	Decreased upon insulin pathway activation [45]	Tyr612			
				Similar upon insulin pathway activation [45]	Tyr896			
NF-kB [41]	Mice	C2C12 myoblasts	NR	Increased [41]	Ser536	Increased binding activity [41]		
PLN [39]	Mice	Heart tissue	NR	Decreased in response to isoproterenol [39]	Ser16			
Rb [24,27]	Human	Myoblasts	NR	Increased [24]	NR			
		Myoblasts to myotubes	NR	Increased at first days of differentiation [24]	NR			
		Neural stem cells	NR	Decreased [27]	Ser801/Ser811			
RpS6 [26,27]	Mice	Muscle tissue	NR	Increased in 24h starvation conditions [26] Similar in fed mice [26] Similar in mice with 45h starvation conditions [26]	Ser235/Ser236 Ser240/Ser244 Ser240/Ser244			
	Human	Muscle tissue	NR	Increased [26]	Ser235/Ser236 Ser240/Ser244			
		Fibroblast-derived myotubes	NR	Increased upon starvation conditions [26]	Ser235/Ser236 Ser240/Ser244			
		Neural stem cells	Similar [27]	Decreased [27]	Ser235/Ser236 Ser240/Ser245			
Stat3 [40]	Mice	Muscle tissue	Similar in mice with moderate muscle wasting [40] Increased in mice with severe muscle wasting [40]	Increased in mice with moderate muscle wasting [40] Increased in mice with severe muscle wasting [40]	Ser727 Ser727			
	Human	Muscle tissue	Similar [40]	Similar [40]	Ser727			
Tau [42,55]	Human Mice PC12 cells	CSF Spinal cord Cell line	Similar [42] Decreased [55] Decreased [63]	Similar [42] Increased [55] Increased [63]	Thr181 Ser396,Ser404, Ser199 and Ser202 Ser396, Ser404 and Thr231			
TSC2 [26]	Mice	Muscle tissue	NR	Similar in fed mice [26] Similar in mice with starvation conditions [26]	Ser1387 Ser1387			
Capzb [54]	Mice	Astrocytes	NR	Increased [54]	Ser263			
Elk1 [60]	Mice	C2C12 myoblasts	NR	Decreased upon MeHg-induced cytotoxicity [58]	Ser383			

Results of similar, increased, and decreased levels are in comparison of DM1 with control (CTL) group. * Absence of comparison with control group. Abbreviations: AKT—Protein kinase B; AMPK—5′ AMP-activated protein kinase; AS160—TBC1 domain family member 4; BB—*Biceps Brachii*; CaMKIIβM—Ca^2+^/calmodulin dependent protein kinase; CDKN1B—Cyclin-dependent kinase inhibitor 1B; CDK4/6/12—Cyclin-dependent kinase 4/6/12; CSF—Cerebrospinal fluid; CTG—Cytosine–Thymine–Guanine triplet; CTL—Control group; CUGBP1—CUG triplet repeat RNA binding protein 1; DMPK—*Dystrophia myotonica* protein kinase; DM1—Myotonic Dystrophy type 1 group; EDL—*Extensor digitorum longus* muscle; eIF2α—Eukaryotic translation initiation factor 2 subunit 1; ERK 1/2—Extracellular signal-regulated kinases 1/2; Fed—Mice food-deprived for 12 h followed by 4 h of free access to food before sacrifice; FOXO1—Forkhead box protein O1; GSK3—Glycogen synthase kinase 3; IGF-1R—Insulin-like growth factor 1 receptor; IkBα—Nuclear factor of kappa light polypeptide gene enhancer in B-cells inhibitor, alpha; INPP4B—Type II inositol 3,4-bisphosphate 4-phosphatase; InsR—Insulin receptor; IRS1—Insulin receptor substrate 1; LBCLs—Lymphoblastoid cell lines; LKB1—Liver kinase B1; MEK—Mitogen-activated protein kinase kinase; MeHg—Methylmercury chloride; mTOR—Mammalian target of rapamycin; n—sample size; NF-kB—Nuclear factor kappa-light-chain-enhancer of activated B cells; NR—Not reported; PDGFRβ—Platelet-derived growth factor receptor beta; PERK—Protein kinase R (PKR)-like endoplasmic reticulum kinase; PKC—Protein kinase C; PKM2—Pyruvate kinase isozyme M2; PKR—Protein kinase R; PLN—Phospholamban; p-protein: phosphorylated protein; PRKAR1B—cAMP-dependent protein kinase type I-beta regulatory subunit; PTPRF—Receptor-type tyrosine-protein phosphatase F; p38MAPK—p38 Mitogen-activated protein kinase; Rb—Retinoblastoma protein; RpS6—Ribosomal protein S6; Stat3—Signal transducer and activator of transcription 3; Ser—Serine; S6K1—Ribosomal protein S6 kinase beta-1; TA—*Tibialis Anterior*; TAK1—Mitogen-activated protein kinase kinase kinase 7; Thr—Threonine; TSC2—Tuberous Sclerosis Complex 2; Tyr—Tyrosine; ULK1—Unc-51-like autophagy activating kinase; 24hStarved/45hStarved—Mice with 24/45 h of food deprivation but free access to water before sacrifice.

## Data Availability

Not applicable.

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
