# Peer review of "Protein Phosphorylation Alterations in Myotonic Dystrophy Type 1: A Systematic Review"

_ijms, 2023, doi:10.3390/ijms24043091_

Round 1

Reviewer 1 Report

This was a comprehensive accounting of the literature regarding protein phosphorylation and myotonic dystrophy.  I think the authors did an excellent job of surveying the literature and compiling tables of results from various papers.  What was missing was some opinion from the reviewers (i.e. judgement).  Which studies did they find more compelling?  Which are more reliable?  Which studies should someone who is new to the field or thinking about entering the field look to as important or more substantive or of higher quality.  When I read a review, I expect that the review was written by experts in the field who will guide me and provide insight into the quality of the various papers.  That is what was missing.  The review was a compilation of data without a statement about which work should be noticed or paid attention to.  

The survey of the literature shows heterogeneous and conflicting results depending on the model system; cells vs animals vs humans.  Variation was even evident depending on which muscle was studied within a species sometimes.  Why?  An expert's insight would be useful.

The most useful elements of the paper were Table 1 and Table S1.  Is there a way to make parts of Table S1 part of the main manuscript?  Table S1 has a nice summary of the data from the literature but the call-out to it is buried at the end of the paper.  IF the authors could draw attention to select papers that they thought were more meaningful throughout their manuscript (cite the paper and refer to it in Table S1 with a highlight of some sort) that would make it more impactful.

I thought the call from the authors to have more studies on phosphatases and the interactions between DMPK and potential targets  was good and an expression of opinion that was welcome.  How about effect on pathways or proteins such as CUGBP1, or maybe even MBNLs?  Could the authors comment on DMPK and it potential impact on some of the other kinases that are mentioned to be altered in their literature review?  Any thoughts, opinions?  This would be interesting since many of the current therapeutic developments for DM1 involve antisense oligonucleotides that target DMPK mRNA and would potentially reduce DMPK levels in treated tissues.

Author Response

Rebuttal Letter

IJMS- 2165341

Reviewer #1

Reviewer 1: This was a comprehensive accounting of the literature regarding protein phosphorylation and myotonic dystrophy.  I think the authors did an excellent job of surveying the literature and compiling tables of results from various papers.  What was missing was some opinion from the reviewers (i.e. judgement).  Which studies did they find more compelling?  Which are more reliable?  Which studies should someone who is new to the field or thinking about entering the field look to as important or more substantive or of higher quality.  When I read a review, I expect that the review was written by experts in the field who will guide me and provide insight into the quality of the various papers.  That is what was missing.  The review was a compilation of data without a statement about which work should be noticed or paid attention to. 

Author Response: Although we understand the reviewer concern, the authors like to mention that the present manuscript is a Systematic Review. Systematic reviews are a type of review that uses repeatable methods to find, select, and synthesize all available evidence to answer a clearly formulated research question. In this type of manuscript, the authors opinion should be avoided and that is the main difference with a review article.

In this manuscript, our main goal was to collect and summarize all the reported alterations related to protein phosphorylation events in DM1. Given the nature of the results (alterations in kinases, phosphatases and phosphoproteins) is very difficult to compare the data obtained for each protein and evaluate all the paper quality. Therefore, we compiled all the alterations already reported, we discussed its possible implications for DM1 pathogenesis and highlight the research gaps in the field that should be explored in future studies, namely the lack of studies evaluating alterations of protein phosphatases.

Moreover, given the amount of information gathered in the selected papers, in this systematic review we discussed and integrated only the findings that we believe that are more robust and that therefore should be more noticed.

Reviewer 1: The survey of the literature shows heterogeneous and conflicting results depending on the model system; cells vs animals vs humans.  Variation was even evident depending on which muscle was studied within a species sometimes.  Why?  An expert's insight would be useful.

Author Response: We would like to thank the reviewer for raising this important issue. In fact, the heterogeneous and conflicting results obtained depending on the model system used is not an unexpected issue. Several in vivo and in vitro biological models have been used to study the molecular mechanisms underlying DM1 with several advantages/disadvantages. For instance, transgenic mice are a biological model of choice that recreates key molecular and phenotypic features such as somatic (CTG)n expansion instability, RNA toxic gain-of- function, abnormal splicing and underexpression of important genes associated to DM1. However, some of the limitations for the use of mouse models are that the levels of repeat instability are decreased in comparison to what is observed in germ lines and somatic tissues in human cell models, possibly due to the use of relatively shorter number of repeats than observed in humans, especially in more severe phenotypes.

Human derived cell models are of great utility for the study of DM1, since they can express the whole range of repeat lengths while staying at their genomic context and reproducing the characteristic features of DM1, such as RNA foci, alternative splicing and metabolic dysfunctions. However, one of the major limitations of primary cells in general, is the limited number of times they can divide until they enter a state of replicative senescence. In DM1, this effect is even more accentuated since cells may enter a state of early senescence compared to the control cells. At the same time, primary cells have great variability between individuals, which alters the growth conditions of the cell line considering age, the tissue of origin and the degree of dysfunction. Therefore, to increase the robustness of the data and integrate the research findings related to DM1, both the use of mouse models and human cell lines should be considered.

Regarding the variations observed in studies performed using human muscle samples, this may be related to the sample collection (i.e., biopsies, post-mortem). Moreover, differences between the type of muscle group, for instance between BB and TA, could be also expected in DM1 given that this muscular dystrophy is characterized by progressive and symmetrical distal muscle weakness.

Given the importance of this issue, we decided to include this idea in the discussion section as follows:

‘Some heterogeneous and conflicting results were reported depending on the model system used. In fact, several in vivo and in vitro biological models have been used to study the molecular mechanisms underlying DM1 with several advantages/disadvantages. For instance, transgenic mice are a biological model of choice that recreates key molecular and phenotypic features of DM1 [82,83]. However, some of the limitations for the use of mouse models are that the levels of repeat instability are decreased in comparison to what is observed in germ lines and somatic tissues in hu-man cell models, possibly due to the use of relatively shorter number of repeats than observed in humans, especially in more severe phenotypes [84]. Human derived cell models are of great utility for the study of DM1, since they can express the whole range of repeat lengths while staying at their genomic context and reproducing the characteristic features of DM1 [85–87]. One of the major limitations of primary cells in general, is the limited number of times they can divide until they enter a state of replicative senescence. In DM1, this effect is even more accentuated since cells may enter a state of early senescence compared to the control cells. At the same time, primary cells have great variability between individuals, which alters the growth conditions of the cell line considering age, the tissue of origin and the degree of dysfunction [88–90]. Therefore, to increase the robustness of the data and integrate the research findings related to DM1, both the use of mouse models and human cell lines should be considered. Further, was also reported variations in the results obtained in studies performed using human muscle samples, which may be related to the sample collection (i.e., biopsies, post-mortem). Moreover, differences between the type of muscle group, for in-stance between BB and TA, are also observed. These differences are in fact expected in DM1 given that this muscular dystrophy is characterized by progressive and symmetrical distal muscle weakness [5]..’

Reviewer 1: The most useful elements of the paper were Table 1 and Table S1. Is there a way to make parts of Table S1 part of the main manuscript?  Table S1 has a nice summary of the data from the literature but the call-out to it is buried at the end of the paper.  IF the authors could draw attention to select papers that they thought were more meaningful throughout their manuscript (cite the paper and refer to it in Table S1 with a highlight of some sort) that would make it more impactful.

Author Response: We appreciate the reviewer comment. In a systematic review is usual to collect all the information relevant for the scientific question in each manuscript and create a table as Table S1 of this manuscript. Given the dimensions, this tables are usually presented as supplementary information.

In this manuscript, in order to organize the data and compile the information of a given protein kinase, protein phosphatase or phosphoprotein, Table 1 was created. It was the information of table 1, which combines the information of all manuscripts collected, that was used to the discussion the results obtained and is then always cited in the results and discussion sections. However, the authors should mention that whenever pertinent, the reference to the manuscripts where the information was collected was also cited in results and discussion.

Reviewer 1: I thought the call from the authors to have more studies on phosphatases and the interactions between DMPK and potential targets was good and an expression of opinion that was welcome. 

How about effect on pathways or proteins such as CUGBP1, or maybe even MBNLs? Could the authors comment on DMPK and it potential impact on some of the other kinases that are mentioned to be altered in their literature review?  Any thoughts, opinions? This would be interesting since many of the current therapeutic developments for DM1 involve antisense oligonucleotides that target DMPK mRNA and would potentially reduce DMPK levels in treated tissues.

Author Response: We thank the reviewer for the very positive comments.

Regarding the effect on pathways or proteins such as CUGBP1 and MBNLs, we in fact discussed this issue all over the manuscript, in particular in subsections 3.5.3 and 3.6. These proteins alterations and impact for DM1 are issues well explored in the field, and in this manuscript, we decide to notice and propose future studies in research gaps.

For DMPK, currently there is no reported evidence suggesting a direct role of this kinase in the other kinases highlighted in this review. DMPK is the DM1 central protein and evidence suggests that due to the occurrence of CUG expanded mRNA that forms hairpin-like structures and accumulates in the nucleus, the DMPK protein levels are diminished. Therefore, although DMPK might not be impacting directly other kinases, is expected that by altering several signaling pathways, some described in this manuscript, it impacts indirectly the expression and activity of other kinases that might be underlying some DM1 heterogenous symptoms. For instance, MBNL1 transcripts containing exon 5 and the respective protein isoforms (MBNL142-43) were found to be overexpressed in DM1 muscle, which bind the Src-homology 3 domain of Src family kinases (SFKs) via their proline-rich motifs, enhancing the SFK activity. This evidence suggests an additional molecular mechanism in the DM1 pathogenesis, based on an altered phosphotyrosine signalling pathway (Botta et al. 2013). In this way, the novel therapeutic developments for DM1 involving oligonucleotides that target DMPK mRNA would be beneficial and are a promising approach to cure the disease. However, given the information gathered in this manuscript regarding phosphatases, kinases and phosphoproteins alterations that as mentioned above might be responsible for the development of some DM1 heterogenous symptoms, we also propose the modulation of these potential novel targets of therapeutic interest for DM1.

Given the importance of this issue, we decided to include this idea in the discussion section as follows:

‘Currently, there is no reported evidence suggesting a direct role of DMPK in other kinases and phosphatases highlighted in this manuscript. Although DMPK might not be impacting directly other kinases and phosphatases, is expected that by altering several signaling pathways, it impacts indirectly their expression and activity which might be underlying some DM1 heterogenous symptoms.’

Reviewer 2 Report

In this amenable to read, well-structured-illustrated, and updated  systematic review (based on carefully selected 41 articles), Costa el al  summarizing the very complex altered protein phosphorilation processes that underlying the clinical manifestations of myotonic dystrophy type 1 (DM1).

Their results identified twenty-nine kinases, three phosphatases and seventeen phosphoproteins linked to the pathogenesis of DM1. These proteins participate into the regulation of cell cycle, cell proliferation and differentiation, apoptosis, autophagy,  glucose metabolism, and stress response, mainly through their alterations into the AKT/mTOR, MEK/ERK, PKC/CUGBP1, AMPK, amomg others cellular patways, disovered in affected biological samples either from patients or cellular/animal models for DM1. Very interestinlgy, these alterations could explain the increased risk seen in DM1 for diabetes mellitus and neoplasms. 

Their findings are well organized, illustrated, described, sumarized and properly cited in Table 1.

Section 3.5 briefly describe the clinical/cellular correlations of the main altered cellular pathways.

This review also emphasizes that understanding the alterations on kinases, phosphatases and phosphoproteins in DM1, could warrant the development of future pharmacological treatments (exemplified by reduction of myotonia and muscle wasting through modulation by AMPK/mTORC1 pathway by the use of rapamycin and AICAR).

I have only the following minor comments: 

When cited for first time, please include OMIM ID entries for all cited human Mendelian disorders (MIM#) and their corresponding responsible genes (MIM*) .

Lane 617: If possible, please specify the concrete increased risk accordingly to the neoplasm type observed for those DM1 patients.

At section of Material&Methods: Please include a brief description of the employed Biorender.com software for creating the very useful and illustrative figure 6. Also, cite the web site for this computational program or its corresponding reference.

Author Response

Rebuttal Letter

IJMS- 2165341

Reviewer #2

In this amenable to read, well-structured-illustrated, and updated systematic review (based on carefully selected 41 articles), Costa et al summarizing the very complex altered protein phosphorylation processes that underlying the clinical manifestations of myotonic dystrophy type 1 (DM1).

Their results identified twenty-nine kinases, three phosphatases and seventeen phosphoproteins linked to the pathogenesis of DM1. These proteins participate into the regulation of cell cycle, cell proliferation and differentiation, apoptosis, autophagy, glucose metabolism, and stress response, mainly through their alterations into the AKT/mTOR, MEK/ERK, PKC/CUGBP1, AMPK, among others cellular pathways, discovered in affected biological samples either from patients or cellular/animal models for DM1. Very interestingly, these alterations could explain the increased risk seen in DM1 for diabetes mellitus and neoplasms. 

Their findings are well organized, illustrated, described, summarized and properly cited in Table 1.

Section 3.5 briefly describe the clinical/cellular correlations of the main altered cellular pathways.

This review also emphasizes that understanding the alterations on kinases, phosphatases and phosphoproteins in DM1, could warrant the development of future pharmacological treatments (exemplified by reduction of myotonia and muscle wasting through modulation by AMPK/mTORC1 pathway by the use of rapamycin and AICAR). 

Author Response: We thank the reviewer for the very positive comments about our manuscript.

I have only the following minor comments: 

Reviewer 2: When cited for first time, please include OMIM ID entries for all cited human Mendelian disorders (MIM#) and their corresponding responsible genes (MIM*).

Author Response: We thank the reviewer for the suggestion which we incorporated in the revised manuscript.

Reviewer 2: Lane 617: If possible, please specify the concrete increased risk accordingly to the neoplasm type observed for those DM1 patients.

Author Response: We appreciate the reviewer suggestion, however we decide to not include this information in the manuscript discussion. In fact, there are a few manuscripts, cited in lane 617 (references 106, 107 and 108), which evaluated the increased risk of cancer development by DM1 patients accordingly to the neoplasm type. However, the information reported in the manuscripts are not compliant, and then we decide to not add this discussion to the manuscript since is not essential to comprehend our results.

Reviewer 2: At section of Material&Methods: Please include a brief description of the employed Biorender.com software for creating the very useful and illustrative figure 6. Also, cite the web site for this computational program or its corresponding reference.

Author Response: The following sentence was included in Material&Methods section:

“BioRender is a web-based tool (https://biorender.com) with thousands of premade icons allowing the creation of professional scientific figures, and was used to create Figures 4, 5 and 6.”

Round 2

Reviewer 1 Report

The authors have addressed my main concerns adequately.